# Exploring the Perceptions of Women from Under-Resourced South African Communities about Participating in a Low-Carbohydrate High-Fat Nutrition and Health Education Program: A Qualitative Focus Group Study

**DOI:** 10.3390/nu12040894

**Published:** 2020-03-25

**Authors:** Georgina Pujol-Busquets, James Smith, Kate Larmuth, Sergi Fàbregues, Anna Bach-Faig

**Affiliations:** 1Division of Exercise Science and Sports Medicine, Department of Human Biology, Faculty of Health Sciences, University of Cape Town, Cape Town 7700, South Africa; ja.smith@uct.ac.za (J.S.); kateus65@gmail.com (K.L.); 2Faculty of Health Sciences, Universitat Oberta de Catalunya (Open University of Catalonia, UOC), 08018 Barcelona, Spain; 3Department of Psychology and Education, Universitat Oberta de Catalunya (UOC), 08018 Barcelona, Spain; sfabreguesf@uoc.edu; 4FoodLab Research Group (2017SGR 83), Faculty of Health Sciences, Universitat Oberta de Catalunya (Open University of Catalonia, UOC), 08018 Barcelona, Spain; abachf@uoc.edu; 5Food and Nutrition Area, Barcelona Official College of Pharmacists, 08009 Barcelona, Spain

**Keywords:** nutrition education, thematic analysis, focus group discussion, low-carbohydrate high-fat, South Africa, under-resourced communities

## Abstract

Scientific evidence suggests that low-carbohydrate high-fat (LCHF) diets may be effective for managing non-communicable diseases (NCDs). Eat Better South Africa (EBSA) is an organization that runs LCHF nutrition education programs for women from low-income communities. Three focus group discussions (FGDs) were held with 18 women who had taken part in an EBSA program between 2015 and 2017, to explore their perceptions and to identify the facilitators and barriers they faced in implementing and sustaining dietary changes. Thematic analysis of the focus groups was conducted using NVivo 12 software. Women reported that they decided to enroll in the program because they suffered from NCDs. Most women said that the EBSA diet made them feel less hungry, more energetic and they felt that their health had improved. Most women spoke of socioeconomic challenges which made it difficult for them to follow EBSA’s recommendations, such as employment status, safety issues in the community, and lack of support from relatives and doctors. Hence, women felt they needed more support from EBSA after the program. The social determinants that affected these women’s ability to change their health behavior are also NCD risk factors, and these should be assessed to improve the program for other communities.

## 1. Introduction

Diet-related non-communicable diseases (NCDs) pose a huge burden in terms of financial cost, morbidity, and mortality [1,2,3]. In South Africa there is currently a double burden of infectious diseases and NCDs [4]. Even so, NCDs in South Africa contribute an estimated 43% of total deaths for all ages and both sexes [5]. Overconsumption of simple sugars, refined carbohydrates, and poor-quality refined oils are believed to increase the risk of developing NCDs [4,6]. Furthermore, there is now much clinical evidence that dietary carbohydrate restriction, and particularly sugar restriction, can improve markers of metabolic health, including obesity, hyperlipidaemia and hyperglycaemia [7,8,9]. Current guidelines for a healthy diet in South Africa recommend carbohydrate intake to be 45–65% of calories and fat intake to be 20–35% of calories [10]. In contrast, low-carbohydrate diets usually include 5–26% of carbohydrate (20–130 grams/day of carbohydrates) and fat intake is often 40–70% of calories [11,12]. Low-carbohydrate high-fat (LCHF) is a term that is often used to describe these diets; however absolute fat intake is often increased only marginally, when total calorie intake is reduced in weight loss versions of this diet [13,14,15]. LCHF diets as a means for weight loss and to improve metabolic health, entered the public sphere in the early 1970s in the form of the Atkins Diet, but became much more popular from the late 1990s [12,16,17]. In South Africa, there is currently much public interest in LCHF diets, to the extent that restaurants have ‘Banting’ options in their menus [17,18,19]. The term Banting refers to an LCHF lifestyle that has severe carbohydrate restriction as its core principle, but also restricts industrially produced seed oils, and promotes consumption of fats that are considered healthy by proponents of this diet, including olive oil, coconut oil and animal fat. Total calorie intake is often regulated by eating to satiety and not beyond, rather than actively restricting calories. Indeed, the diet is mostly promoted as one which increases control of eating and satiety [20,21,22]. 

There is a general perception that the foods included in an LCHF diet are expensive and that most of the people who decide to follow this lifestyle in South Africa are reasonably affluent [22]. This is consistent with the perception that LCHF diets are too expensive for people living in under-resourced communities [23]. However, it is people from disadvantaged communities that suffer the highest rates of illness from NCDs due to limited access to education and health care [24,25,26]. As such, vulnerable populations from low socioeconomic communities in South Africa often remain undiagnosed and/or untreated, which increases their risk of complications from NCDs [5,24,26]. South Africa remains a dual economy and, according to a 2018 report by the World Bank, has the highest inequality rate in the world [27]. Inequality in South Africa is strikingly apparent with affluent suburbs surrounded by vast townships. The term township refers to the often underdeveloped and racially segregated urban areas that were originally reserved for non-white inhabitants during the Group Areas Act (1960 to 1983), which mandated residential racial segregation under the Apartheid government [28,29,30]. Consequently, black, mixed-race ancestry, and people of Indian/Malay descent were forcefully removed from designated white areas to areas that were often far from places of employment in the central cities [30]. Despite an end to the Group Areas Act, racial inequalities have persisted in economic form and in access to healthcare and education. In addition, the disadvantages that women face in South Africa constitute a major source of inequality [31,32]. Therefore, women living in these townships are confronted with the discriminatory double-edged sword of poverty and gender inequality [31,33].

There are several nutrition education programs that operate in low socioeconomic settings in South Africa in an attempt to reduce the burden of NCDs [34,35,36]. Eat Better South Africa (EBSA) is an organization which was founded in 2015 to address health related inequality issues [37]. To date, EBSA has run several dietary education programs in under-resourced, predominantly mixed-race townships. The aim of EBSA is to empower under-resourced people—particularly women—to improve their health by making the best dietary choices available to them. EBSA programs are 6 weeks long and involve weekly 2-h education sessions to teach participant about nutrition, NCDs, shopping on a budget, cooking and eating healthier foods. Participants also join an online instant messaging group (WhatsApp, Mountain View, CA, United States) where they can ask questions and share motivation with each other and the EBSA team. EBSA’s recommended diet is based on the LCHF/ Banting lifestyle. The primary focus is to reduce the consumption of simple sugars, refined carbohydrates, refined seed oils, and ultra-processed junk food. Foods that are recommended include affordable sources of vegetables, eggs, dairy, fish, chicken, and red meat (including organ meat) [37]. The education sessions are run by an EBSA educator with the help of an EBSA ambassador and coaches. EBSA ambassadors are usually public figures that can provide motivation to improve health during the program. EBSA coaches are women hired from the local communities to manage the participants’ WhatsApp group and to organize follow up meetings as a means to encourage sustainability of the diet. 

As far as we are aware, no study has investigated the perceptions and experiences of participants taking part in an LCHF dietary intervention in South African communities where socioeconomic challenges such as poverty, unemployment and crime are prevalent. With an absence of evidence-based information, organizations that provide LCHF education in these communities, lack proper insight to evaluate and improve future programs. The current qualitative study therefore examines the perceptions and experiences of women who had taken part in an EBSA program, and by inference, had attempted to follow an LCHF diet. The purpose was to understand whether an LCHF diet was accessible and affordable to members of these communities, and to identify ways to improve the acceptance, compliance, and sustainability of future EBSA programs. 

## 2. Materials and Methods 

### 2.1. Study Design and Ethical Approval

This was a cross sectional qualitative study that used focus group discussions in women who had previously taken part in an EBSA intervention. Ethical approval (HREC REF 391/2018) was granted by the Faculty of Health Science Human Research Ethics Committee of the University of Cape Town and written informed consent was obtained from all participants. The study was conducted in accordance with the Declaration of Helsinki.

### 2.2. Study Settings

This study included women from Ocean View who had taken part in an EBSA intervention in June 2015 (O15) and women from Atlantis who had taken part in an EBSA intervention in June 2016 (A16) or June 2017 (A17). Both communities are in the Western Cape Province of South Africa and originated during the Group Areas Act under the Apartheid government [28]. Ocean View has a population of nearly 14,000 people and around 3083 households. 51% of the community are female and 91% are from mixed-raced ancestry. Afrikaans is the most used language and is the first language for 57% of the population, followed by English at 39% [38,39]. The Ocean View program of 2015 was led by the EBSA educator who was also a famous South African actress. All of the women who took part in this program were already part of a wellness group that met every week to exercise at a multi-purpose hall. Atlantis was established as an industrial center and as a residential area for those of mixed-raced ancestry. There are close to 68,000 people living in Atlantis with around 15,565 households. Approximately 51% of the community are females and 85% are from mixed-race ancestry. Afrikaans is also the most used language, spoken as the first language by 80% of the population, followed by English at 9% [38,39]. The 2016 Atlantis EBSA program was led by the same EBSA educator (famous actress) who led the Ocean View program, but the 2017 Atlantis program was led by another EBSA educator. The 2017 Atlantis EBSA coach was appointed after taking part in the 2016 Atlantis intervention and demonstrating potential for this role.

### 2.3. Participants

The coach and ambassador from previous EBSA programs helped to recruit eligible participants for this study, using either the EBSA WhatsApp group, by telephone or at one of EBSA’s follow up meetings. Purposive criterion sampling [40] was used to select participants fulfilling the following eligibility criteria: women; 18 years or older; capable of providing informed consent; able to understand and speak English or Afrikaans; able to attend the focus group meeting; and must have attended at least two sessions of the EBSA program. Women who were interested in taking part in the study were given a participant information sheet and those who met the eligibility criteria were recruited as part of the study sample.

### 2.4. Procedures

Focus group discussions (FGDs) were conducted between September and October of 2018. The FGDs included 5 to 7 women and lasted, on average, 1 h and 50 min. In accordance with recommendations in the literature, our relatively homogeneous sample was large enough to identify 90% of the common themes [41,42]. The group discussions were held in a venue within the community. Upon arrival, each participant reviewed and signed an informed consent and completed a sociodemographic questionnaire in the presence of a member of the research team who was fluent in the participant’s preferred language (English or Afrikaans). A bilingual (Afrikaans and English) woman, who was independent of EBSA, was the focus group discussion moderator. Participants were asked to express themselves in the language they felt most comfortable in, which was predominantly Afrikaans. The moderator encouraged conversation among participants using ten open-ended questions, which were developed by the researchers for this study and were not necessarily asked verbatim (Appendix A). Prompts and probes were used when necessary, to elicit further information. During the focus group, a researcher who assisted the moderator, wrote notes which were used to create a context and give clear and consistent details of the discussion. Refreshments were provided at each focus group and participants were reimbursed as compensation for their time.

### 2.5. Data Analysis

The focus groups discussions were audio recorded, translated, and transcribed from the original language (mostly Afrikaans) into English. Using NVivo 12, thematic analysis [43,44,45] was conducted on the translated transcripts. Thematic analysis involved six phases [43]. The first phase involved GPBG reading the transcripts several times and identifying patterns in the data. In the second phase, a first draft of the codebook containing 60 codes was developed by GPBG based on the previous readings. Subsequently, ABF and SF revised and abbreviated the codebook by excluding 19 codes, leaving a total of 41 codes. The third phase involved GPBG searching for themes by structuring each code into several sub-themes. In the fourth phase, sub-themes were reviewed by ABF and SF, being subsequently displayed by GPBG as thematic maps. The fifth phase involved naming the themes and defining the main concepts of each theme, resulting in a total of six themes. The last phase, mainly carried out by GPBG, KL, and JS, consisted of producing a report which was shared with the EBSA team to optimize the program for future interventions. Although only GPBG did the coding, which has the potential to introduce personal bias, all the researchers had reviewed the original transcripts and coded data, and agreed on the final themes and their interpretation in this article.

## 3. Results

A total of 18 women completed the study by taking part in one of three FGDs. All participants reported that they had mixed-race ancestry. Mean age was 51.7 ± 11.9 years and mean body mass index (BMI) was 30.1 ± 6.7 kg/m^2^. Table 1 shows other participant characteristics that were reported in the study questionnaire.

Six themes were developed from the analysis: (i) motivations to enroll in the EBSA program; (ii) perceptions of the EBSA program structure; (iii) experiences with the recommended diet; (iv) EBSA’s influences on participants’ lives; (v) challenges during and after the EBSA program; and (vi) suggestions to improve the EBA program. Table 2 shows the themes, sub-themes, and the definition of the theme. 

### 3.1. Theme 1: Motivations to Enroll in The EBSA Program

Improving health was the major driver to participate in the EBSA program. Most of the women said they had pre-existing metabolic conditions, and many mentioned that they wanted to lose weight. Women acknowledged concerns around their health problems and anxiety that their unhealthy eating habits would lead to their premature death. Overall, the participants voiced a great desire to change their lives and become healthier.

“I was very fat and a diabetic. I had high blood pressure and cholesterol. (A16)” 

“I cannot die before the time by not eating right. (O15)” 

“I decided that I am also going to join because I want to make a change in my life. (A17)”

This was further emphasized by the participants´ references to family health history as their reason to take part in the program, as some had relatives who were dying or had died from metabolic conditions.

“We have a family history of diabetes. I was told that I needed monitoring. (…) I decided to start Banting (A17).”

Participants from A17 mentioned that they were encouraged by other women from the community who had participated in EBSA programs the year before. Participants from O15 and A16 said that they joined EBSA because it was offered to their wellness group for free. Some women said that they were influenced by groups on social media that promoted an LCHF lifestyle. Participants were particularly motivated by people’s pictures from before and after going on the “Banting” diet. 

“One of my friends joined (the EBSA program), she experienced an improvement in her health and weight, and that is the reason why I joined. (A17)”

### 3.2. Theme 2: Perceptions of The EBSA Program Structure

Women said that being part of a cohesive group of women was one aspect that they valued most from the program. In fact, it was mentioned that motivation dropped if the others were not attending the education sessions. In terms of EBSA’s team support, women from the A17 program reported being very happy with their coach and her continued support through the WhatsApp group, suggesting she was key to their experience.

“Just to know that there are other people with you [in the EBSA program]. If you are on your own you could easy quit, but if you are with other people you feel more motivated. (O15)” 

“One thing that put me off was the [lack of] attendance of the people. (A17)”

“The coach is very hands on with the WhatsApp group. Therefore, if we have any questions, we can go to the WhatsApp group and ask her directly. (A17)” 

Generally, participants spoke favorably of the educators. However, participants from A17 complained that the educator did not speak Afrikaans, and they found this was not inclusive enough for the mostly Afrikaans speaking community. The delivery of the educational sessions was perceived as very informative and helpful by some women; however, there was one complaint about some of the education material being too academic. In addition, some women felt uncomfortable about the lack of privacy when undergoing body measurements and the competitive nature of the weigh-ins which were intended to motivate weight loss during the program.

“The educator could not speak Afrikaans fluently. (…) Many of the people in my area have had a huge problem with English. (A17)”

“I found it [the EBSA program] to be very informative and I learned things that I never knew about sugar.” (A17)

“They (the EBSA sessions) are academically long with many graphics. (A17)”

“I do not like being in a group where we have to compete and the one looks to see how the other one looks. (A17)” 

Different views were expressed about whether the total number and frequency of the sessions was appropriate: some women mentioned that they were happy with the six sessions; some would have liked more sessions, while others would have preferred a larger gap between the sessions as they found the six sessions in a row were too much of a commitment

“We had the six meetings, which means it was six Saturdays consecutively and I feel that there should have been a gap. (A17)”

### 3.3. Theme 3: Experiences With the Recommended Diet

According to almost all of the women, the meal plans provided during the programs, were easy to follow and very helpful. One woman mentioned that this diet reminded her about how her ancestors used to eat. In general, women said that the diet made them feel more energetic, satiated for longer and that they could eat without feeling guilty. 

“What makes more sense for me regarding the program is how our ancestors ate and how my grandmother prepared food. (…). Everything was full cream. There weren’t low fat products and skimmed milk. (A16)”

“I felt energetic and I did not feel sleepy. My thoughts were sharper. I was more alive. (A16)” 

“It [the LCHF diet] keeps you full for longer. (O15)” 

“You can now eat without feeling guilty. (A16)” 

Foods that many participants reported eating included eggs, cream, coconut oil, non-starchy vegetables, and chicken. Eggs in particular were highly regarded by most participants as an important and versatile food. Participants expressed an appreciation that eating certain fats like butter and animal fat was permitted more freely with the EBSA diet and said that this improved the taste of their food. Adding coconut oil or butter to coffee (Bulletproof coffee) as a meal substitute was reported quite often.

“I think eggs are the most versatile out the whole diet. (O15)”

“It is so nice to buy a piece of fat and put it over the vegetables. (A17)”

“About a teaspoon of coconut oil, a dash of cream, a dash of milk. (…) That is the bulletproof coffee. (O15)”

Participants were specifically asked about foods that EBSA had recommended that they disliked. Most participants were not dissatisfied with any of the foods except the taste of a coconut and seed-based flour product which was promoted as a substitute for grain-based flour. 

“The porridge [low-carbohydrate version] I do not really like, it does not have a great taste [laughing], it puts you completely off, huh. (A17”)

Participants admitted that it was quite difficult to give up certain foods that were excluded from the EBSA diet, particularly processed foods such as chips, chocolate, and sugar sweetened beverages. 

“I love chocolate… [laughing] Yes, that was actually difficult for me to get rid of. (O15)”

Many participants said that they could afford to follow the EBSA diet during the program and that it was not too expensive for them. There were some participants that said that they struggled to afford the recommended foods after the program, particularly those that were unemployed or were retrenched. Another unexpected factor that some women experienced around food was that certain food items that would ordinarily be available in local shops, were difficult to get hold of. Some participants felt that these items would sell out quickly due to the increased demand by EBSA participants, who were following the same meal plans. In terms of grain substitutes, women stated that the coach played an intermediary role by supplying ‘Banting’ products promoted by EBSA, which were not usually available in their communities.

“It wasn’t a problem [the diet]. I could make anything, and I did not spend a lot of money because it was natural stuff. Vegetables, eggs and chicken. (O15)”

“I wanted to carry on [Banting], but then I was retrenched. There was no money to carry on. (O15)” 

“Sometimes the shop does not have butter or double cream yogurt because everybody wants to be on Banting. (A16)”

“The coach provides Banting food [products labelled as……“Banting”], people can order from her. (A17)”

### 3.4. Theme 4: EBSA’s Influences on Participants’ Lives

Overall, it appears as though the EBSA program had a positive impact on participants’ lives as they said it provided many health benefits. Some participants stated that they became more active and started to exercise as a direct result of the program. Women reported that they felt healthier and they were no longer using some medication (e.g. antihypertensives).

“The more I go, the more educated I am going to become and the more [weight] I am going to lose. (O15)”

“When I started with the program I didn’t really like to exercise. When the Banting program started, SC motivated me to walk the first 10km. From then on, I do every walk that there is. (O15)”

“I no longer needed my high blood pressure pills. (A17)”

According to some of them, one factor which made the biggest impact was the nutritional knowledge they gained from the program. Understanding the harmful consequences of sugar and refined carbohydrate consumption was something they said they will never forget. In addition, women mentioned that there were other people who would benefit from having this knowledge and were encouraging them to join new programs or to try the diet.

“When you go shopping, you will always be checking the ingredients (…). Subconsciously you know what you are allowed to and what you are not. You would always have that in the back of your mind (…). So, there is always that filter. (O15)” 

“When you look at people that is terribly overweight, or people eat things that they are not supposed to, then you intervene immediately. (A17)”

### 3.5. Theme 5: Challenges During and After The EBSA Program

When asked about some of the challenges they faced with regard to participating in the EBSA program, the most notable one was the lack of support from people they interacted with. Many participants complained that they were not supported by the medical practitioners they saw, because they did not agree with the EBSA recommended diet. Additionally, participants felt a lack of support from friends, relatives, and work colleagues during the early stages of the program. However, some participants reported that they received more support from their doctors and peers when they saw their health improvements. 

“When I went to see my physician (…) I told her I am Banting. She said I must not do this because it is a lot of fat and I must stop it immediately. When I went to her again after three months, I sat there all quiet and I did not say to her I am still on Banting. However, she said to me whatever you are doing just go ahead. (A17)” 

“I absolutely got no support. Not from my husband, not my mother, from nobody in the beginning. (…) When they saw what the program did to me and how I was (…) they began to support me. (A17)”

A major challenge experienced by participants from O15 was the gang violence in the area. Women mentioned that lack of safety prevented them from being more active and made them feel stressed and anxious. 

Some women who lost a lot of weight during the program complained of an unexpected expense because they had to buy new clothes. 

“I think that the disadvantages were my clothes, it did take a lot of money. My clothes immediately were bigger. (A17)”

Poor weather conditions appeared to be a challenge for some women to engage in activities and be motivated.

“It was winter [during the EBSA program] and it rained. Believe me the weather does play a role. (A16)”

### 3.6. Theme 6: Suggestions to Improve the EBSA Program

When asked about ways in which the EBSA program could be improved or optimized, women made many suggestions. A major one was that the follow up sessions required a more structured schedule because attending these was seen as an important facilitator to adhering to the recommended diet. Also, participants mentioned that they did not know what to do once they reached their weight goals, so they proposed that EBSA should monitor them and provide weight maintenance plans. Some women even suggested that EBSA should run more than one program within the same year in the same community. 

“For instance, there are no fixed dates; (for the follow up sessions); (…) However, if we could come together and put fixed dates, then we know I need to make time to attend because support is very important. (A17)”

“We want to maintain (weight). (…) They need to provide a session on maintenance. (A17)”

“It the (EBSA program) should be twice a year [laughing]. (…) With a gap in between. (…) And during that big gap you start the monitoring. (A16)”

Women felt that EBSA’s message would be better spread to the wider community if EBSA would make use of the local newspaper or radio station. Moreover, they mentioned that providing before and after pictures of participants who took part in the program would be a good motivator for women interested in the EBSA program. Participants also suggested that EBSA should run similar educational programs within the school environments or hospitals as they felt they would benefit from the nutritional knowledge.

“We have a radio station, (…) we can get somebody to talk about healthy eating. (A17)” 

“The children are young, and they are supposed to learn the right way of eating. This could have a ripple effect and all these chronic diseases. (…) If we can do some awareness about it especially among children, it will also help. (A17)”

“You will see all the people sitting there at the day hospital. They, young and old, are diabetics, they are obese and have high blood pressure. I have been taught the do’s and don’ts in terms of eating. (…) However, this is something new and this needs to be introduced to the broader community. (A17)”

## 4. Discussion

This study explored the experiences of a sample of women who had participated in an EBSA education program that promoted an LCHF diet as a sustainable lifestyle to prevent and treat metabolic health conditions. These participants were from two under-resourced South African communities where residents have limited access to education, employment, and health care [30]. Alcohol and drug abuse, violent crime and teenage pregnancies are also prevalent within these communities [30]. To our knowledge, this is the first study to document women’s experiences with an LCHF diet in a challenging socioeconomic environment in South Africa. Given that LCHF diets are perceived as expensive, and that they conflict somewhat with current dietary guidelines, one purpose of this study was to understand whether LCHF diets would be considered culturally acceptable and affordable among women from these communities. A second purpose was to identify ways to improve the EBSA program.

A major finding was that the EBSA program and its recommended diet was well understood and well received by these participants. It is important to note however that the recruitment of participants in this study would likely have favored those that had an overall positive experience with the EBSA program. Any participants who were very dissatisfied with EBSA or an LCHF diet would probably have been more difficult to reach and less likely to enroll in the study. It is also unclear whether participants were actually following an LCHF diet or whether their body composition and health had in fact improved. Nevertheless, the very positive attitude of these participants toward an LCHF diet is encouraging. An interesting finding was that participants favorably related an LCHF diet to the way that their ancestors ate. This was also reported in a study with Māori participants [46], suggesting that an LCHF diet was easily accepted in those individuals [46]. As is common with other dietary intervention studies [47,48,49,50], the desire to improve health and lose weight were important reasons for these participants to enroll in the EBSA program. Experiencing their own health and weight improvements during the program and being able to achieve this with less hunger and more energy than expected, was likely a strong motivating factor for participants to so convincingly accept an LCHF diet and to adhere to it. That LCHF diets can increase control overeating, through reduced hunger and cravings has been well documented before [20,22,51]. Dietary intervention programs should however be cognizant of the way they measure and use individual health parameters as motivation, such that participants feel comfortable. Being part of a cohesive and supportive group where all participants were following the same diet was another factor that motivated these participants to attend sessions and adhere to dietary changes, and this has been documented in other studies [52,53]. The use of an online group messaging application, in this case WhatsApp, appeared to be instrumental in facilitating this group support outside of the scheduled sessions. Indeed, internet-based communication technology is becoming more valued as a tool for modifying behaviour in nutrition interventions, as it allows on demand communication with a large number of participants [14,20,53,54].

Family support is believed to be of aid in adherence to a range of dietary and health interventions [55,56,57,58]. In contrast, participants in this study identified the lack of support from family and peers as a major challenge during the EBSA program and it made eating in social situations difficult. Similarly, the lack of support, and conflicting advice from doctors caused confusion and frustration among participants. These experiences with peers and doctors are remarkably similar to that of a much more affluent group of individuals with type 2 diabetes who had also followed an LCHF diet [22]. In both studies, family, peers and/or doctors became more supportive of the participants’ efforts after observing their health and weight improvements. LCHF diets remain highly controversial, particularly because of the health concerns related to increased fat consumption and reduced whole grain consumption [59]. An important avenue for future research is to gain a clear understanding of how community doctors perceive an LCHF diet and it is also important for LCHF dietary interventions to engage with community doctors in an attempt to provide consistent messages to participants.

A qualitative study conducted in New Zealand reported that financial constraints prevented participants from adhering to an LCHF diet for an extended period of time [46]. While this was the case for some participants in our study, most participants said they could afford the foods on the program’s meal plan. A limitation of our study was that neither diet nor socioeconomic status was evaluated quantitatively in these participants. It is, therefore, not clear to what degree this sample represents the financial status of all EBSA participants and their wider communities. Although they all lived in low income communities, it is possible that they represented a more affluent part of the community. It is also possible that financial constraints may have prevented other women from joining the EBSA program or following the diet. Nevertheless, these findings suggest that cost effective LCHF meal plans are affordable, at least by some residents of low income communities. In congruence with other studies in low income communities, social factors such as gang violence were reported as a major challenge to follow a healthy lifestyle [5,30,33,46]. Unfortunately, gangsterism, drug abuse and violent crime are common in townships around Cape Town [30,60]. While dietary intervention programs are unlikely to have the means to change these social issues directly, they should be aware of the limitations that they place on participants. These include high levels of social stress and a difficult environment to conduct physical activity.

## 5. Conclusions

The major finding from this study was that the EBSA program and its recommended LCHF diet was generally well understood and well received by these participants. Additionally, participants believed that their dietary changes had led to health improvements. This study highlighted some aspects of the program that participants felt were important to its success, for example the cohesive support group, as well as some challenges that need to be addressed, such as more structured follow up support. Information from this study, together with a site-specific community assessment was used to improve the design and delivery of a subsequent EBSA program. This program will be evaluated longitudinally using both qualitative and quantitative assessments to address some of the limitations of the current study.

## Figures and Tables

**Table 1 nutrients-12-00894-t001:** Participant characteristics (*n* = 18).

Characteristics	*n* (%)
**EBSA program**	
Ocean View 2015 (O15)	5 (27.8%)
Atlantis 2016 (A16)	2 (11.1%)
Atlantis 2017 (A17)	11 (61.1%)
**Program Attendance (6 sessions)**	
2–3 sessions	10 (55.6%)
4–5 sessions	3 (16.7%)
6 sessions	5 (27.8%)
**Highest level of education completed**	
Primary School	1 (5.6%)
High School	13 (72.2%)
Certificate	2 (11.1%)
Diploma	1 (5.6%)
Degree	1 (5.6%)
**Work status**	
Retrenched	1 (5.6%)
Retired	5 (27.8%)
Looking for a job	1 (5.6%)
Employed	11 (61.1%)
**Reported medical conditions**	
High blood pressure	12 (66.7%)
High cholesterol	6 (33.3%)
Obesity	3 (16.7%)
Type 2 Diabetes Mellitus	2 (11.1%)
Heart problems	1 (5.6%)
Asthma	1 (5.6%)
Glaucoma	1 (5.6%)
Psoriasis	1 (5.6%)
None	4 (22.2%)

**Table 2 nutrients-12-00894-t002:** Themes developed from the focus groups.

Theme	Definition of The Theme	Sub-Theme
Theme 1: Motivations to enroll in the EBSA program	Participants’ intentions when deciding to take part in the EBSA program	1.1. Conscious about health problems 1.2. Desire to be healthier 1.3. Worrying about family health history 1.4. Influenced by something or someone
Theme 2: Perceptions of the EBSA program structure	Participants’ opinions about the EBSA program’s structure, sessions, and format	2.1. Group support 2.2. Participant’s attendance 2.3. WhatsApp group 2.4. Coach’s support 2.5. Educational sessions
Theme 3: Experiences with the recommended diet	Participants’ opinions with the LCHF diet and its recommended foods	3.1. Meal plans 3.2. Feelings around diet 3.3. Common foods 3.4. Foods disliked 3.5. Foods missed 3.6. Affordability 3.7. Availability
Theme 4: EBSA’s influences on participants’ lives	EBSA’s impacts on participants’ lives during and after the program	4.1. Health benefits 4.2. Nutritional knowledge 4.3. Encouraging other people
Theme 5: Challenges during and after the EBSA program	Difficulties that EBSA participants encounter during and after the program	5.1. Lack of support 5.2. Safety issues 5.3. Unexpected expenses 5.4. Weather conditions
Theme 6: Suggestions to improve the EBSA program	Participants’ ideas to optimize the program and spread the word in broader the community	6.1. Lack of follow-ups 6.2. Weight maintenance 6.3. Program frequency 6.4. Broader community

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
