# Peer review of "Exploring the Perceptions of Women from Under-Resourced South African Communities about Participating in a Low-Carbohydrate High-Fat Nutrition and Health Education Program: A Qualitative Focus Group Study"

_nutrients, 2020, doi:10.3390/nu12040894_

Round 1
Reviewer 1 Report
I appreciate the opportunity to review this paper, which addresses an important topic of relevance to this and other low- and middle-income countries, i.e., the large prevalence of overweight/obesity and non-communicable diseases, and the effort to address it by improving the quality of the diet.
The authors have a sound theoretical approach to the topic under research, and present their findings in a clear, systematic way. The discussion is well focused and based on the results presented.
My only comment for the authors to consider is whether they would like to highlight the nature of the nutrition education program in their title, to reflect the low-carbohydrate high-protein diet (they actually refer to this term in their Keywords).
Reviewer 2 Report
Very interesting job.The article clearly describes the method used and the purpose of the qualitative research, as well as the results obtained from focus groups.
On the other hand, there is doubt as to the tone of the article from which it can be concluded that the LCHF diet has a beneficial effect on health, despite the fact that the authors do not present such results or cite studies of other authors investigating the impact of the diet promoted by the EBSA.
The respondents' feelings do not provide grounds for assessing the effectiveness of the diet. Changes could have been the cause of many factors: group support, increased physical activity, reduced consumption of sweetened drinks, chips, sweets.
Some examples:
125 All of the women who took part in this program were already part of a wellness group that met every week to exercise at a multi-purpose hall.
201 View 2015 (O15) and Atlantis 2016 (A16), participants said that they joined EBSA because it was offered to their wellness group for free. Some women said that they were influenced by groups on social media that promoted an LCHF lifestyle. Participants were particularly motivated by people’s pictures from before and after going on the “Banting” diet. “One of my friends joined [the EBSA program], she experienced an improvement in her health and weight, and that is the reason why I joined. (A17)”
210 In terms of EBSA’s team support, women from the A17 program reported being very happy with their coach and her continued support through the WhatsApp group, suggesting she was key to their experience. “ Just to know that there are other people with you [in the EBSA program]. If you are on your own you could easy quit, but if you are with other people you feel more motivated. (O15)”
In my opinion, it is worth emphasizing in the article that the diet should be adapted to the needs of a person - based on official dietary recommendations.
Reviewer 3 Report
Title: Exploring the perceptions of women from under-resourced South African communities about participating in a nutrition education program: A qualitative focus group study.
Manuscript ID: nutrients-746331
This is a very well written manuscript. I have some minor comments and a major concern about the lack of information on the qualitative data analysis.
Major comment:
The data analysis section lists important steps but lacks transparency on the process that assure the integrity of the data presented. A clear explanation is needed on the following aspects of data analysis:
- How many researchers were involved in the data analysis process and clearly define what their roles were?
- What was the process/criteria for code elimination to create the final codebook?
- The reliability of qualitative data analysis (e.g. number of coders and the inter-rater reliability of coders)
- Addressing researcher bias. For instance, if a single research was involved in reading the transcripts and coding, then a statement on reflexive bias is needed.
Minor comments:
- Line 155: What was the process for preparation of the focus group questions? Who prepared the questions and how were finalized? It would be informative to provide the questions in the manuscript.
- Line 240: The quote implies a challenge in participating in the intervention due to an external factor (weather) and not the EBSA program structure. It seems to be more fitting to the theme 5 than theme 2. Please provide an appropriate code for theme 2.
